# Research Regarding Dental Mobility Phenomena in the Clinical Recognition Diagnosis of Temporomandibular Disorders

**DOI:** 10.3390/diagnostics13040598

**Published:** 2023-02-06

**Authors:** Laura Elisabeta Checherita, Magda Ecaterina Antohe, Lupu Iulian Costin, Mihai Bogdan Văscu, Ovidiu Stamatin, Irina Croitoru, Sorina Mihaela Solomon, Silvia Teslaru, Irina Gradinaru, Vasilica Toma, Bulancea Petru Bogdan, Daniel Petru Cioloca, Ovidiu Dumitru Aungurencei, Carina Ana Maria Balcoș, Ana Maria Fătu

**Affiliations:** 12nd Dental Medicine Department, Faculty of Dental Medicine, “Grigore T. Popa” University of Medicine and Pharmacy, 16 Universității Street, 700115 Iasi, Romania; 23rd Dental Medicine Department, Faculty of Dental Medicine, “Grigore T. Popa” University of Medicine and Pharmacy, 16 Universității Street, 700115 Iasi, Romania; 3Department of Foreign Language, Faculty of Dental Medicine, “Grigore T. Popa” University of Medicine and Pharmacy, 16 Universității Street, 700115 Iasi, Romania; 41st Dental Medicine Department of Dento-Alveolar and Maxillo-Facial Surgery, Pedodontics Discipline, Faculty of Medicine, “Grigore T. Popa” University of Medicine and Pharmacy, 16 Universității Street, 700115 Iasi, Romania

**Keywords:** diagnostics, periodontal diseases, dental mobility, oral rehabilitation, stomatognathic system, temporo-mandibular disorders

## Abstract

The main objective of this study is to quantify the implications of the complications of periodontal pathology and dental mobility on the pathology of dysfunctional algo syndrome, a clinical entity with profound implications for the patient’s quality of life. Methodology: Clinical and laboratory evaluation was conducted in the 2018–2022 period, on a group of 110 women and 130 men, aged between 20–69, selected from our practice venue, Policlinica Stomatologica nr. 1 Iasi, Clinical Base of Dentistry Education “Mihail Kogalniceanu” Iasi, “Grigore T. Popa” University of Medicine and Pharmacy Iasi and “Apollonia” University Iasi. Overall, 125 subjects were diagnosed with periodontal disease with complications and TMJ disorders and followed periodontal therapy in the context of oral complex rehabilitation treatments (study group); the results of their clinical evaluation were compared with the results of the evaluation of the control group, made from the other 115 patients). Results: Dental mobility and gingival recession were identified as more frequent in the study sample compared with the control sample, the differences being statistically significant in both cases. In total, 26.7% of patients had different types of TMJ disorders and 22.9% of patients had occlusal changes; the percentages are slightly increased in the study sample compared with the control one, but the recorded differences are not statistically significant. Conclusions: Dental mobility, most of the time, is a negative consequence of periodontal disease, leading to the alteration of the mandibular-cranial relations, materializing in an important proportion as an etiopathogenic factor of the dysfunctional syndrome of the stomatognathic system.

## 1. Introduction

Contemporary dentistry has acquired new values and dimensions from the very development of diagnostic concepts and therapeutic decisions, themselves under the influence of modern and complex technologexcept for parts of social psychology and communication The patients are considered in the systemic complexity of their pathological manifestations, as dental conditions often develop in the context of the existence of general conditions, with or without resonance in the dental system [1].

Temporomandibular pathology is a clinical entity affecting more than 25% of the population, with a varied etiology, and a course that is not always predictable. Temporomandibular pathology is a class of musculoskeletal disorders associated with morphological and functional deformities; within temporomandibular pathology, 70% of cases are accompanied by incorrect disc positioning [2].

Assessing the influence of craniomandibular dysfunction on the patient’s quality of life should be an essential element in health policy decision-making. Orofacial pain associated with craniomandibular dysfunction (CMD) is the predominant influence on quality of life, through the impairment of work activity (59.09%), sleep (68.18%) and appetite (63.64%). The psychosocial component of the quality of life is also influenced by CMD, with depression and anxiety being nine times more common in people with CMD than in the unaffected population. It is currently estimated that 75% of the population has at least one sign of craniomandibular dysfunction and approximately 33% have at least one dysfunctional symptom [3,4].

The signs and symptoms of craniomandibular dysfunction increase in frequency and severity between the second decade of life and the fourth. Although most studies show insignificant gender differences in the prevalence of a dysfunctional syndrome, epidemiological studies point to a female/male ratio of 3:1 to 9:1 in people seeking treatment for craniomandibular dysfunction. As outlined, dental malocclusion has been regarded as the main etiologic factor of craniomandibular dysfunction, particularly the occlusal pattern type, occlusal interferences in the centric and maximum intercuspidal relationship and non-occlusal interferences. Malocclusion is associated with complications of periodontal diseases in which dental mobility plays an important role, leading to various complications, materialized by both vertical and horizontal migrations, which disturb the static and dynamic jaw ratios [5,6].

Mobility is defined as the degree of looseness of the tooth. All teeth have a slight degree of physiologic mobility which varies for different teeth and at different times of the day. It is greatest in the morning and progressively decreases [7,8].

Periodontal disease (PD) is generally caused by bacteria in the mouth infecting the tissue surrounding the present teeth [9,10]. Severe periodontitis has been estimated to affect 11.2% of the general population, representing the sixth most prevalent disease worldwide at present. Observational, intervention and experimental studies have associated PD with a number of comorbidities, such as diabetes mellitus, respiratory disorders, atherosclerosis and post-radiotherapy cancer, while recent studies suggest that PD is a risk factor for myocardial dysfunction. Males are affected more often than females [11,12].

Periodontitis, also called gum disease, is an important gum infection that damages the soft tissue and, without treatment, can destroy the bone that supports the teeth. The multiplicity of etiological factors leads to the inclusion of periodontitis in the group with complex multifactorial etiology. Systemic factors are predisposing factors that act by reducing the defense potential of the host tissues or by reducing the functional capacity of the periodontium, which becomes vulnerable to the action of the local factors [13]. Excessive occlusal forces may not cause inflammation, but they lead to degenerative changes in the structure of the deep periodontium, thus causing the inflammatory process of the gingival tissues to expand much more rapidly, leading to much greater destruction of periodontal tissues. Nutritional imbalances and deficiencies can amplify the harmful effects of local irritants and interfere with disease progression. Dietary imbalances influence the response of the periodontium to local irritants, infection, and tissue repair [14].

The main objective of the study is to quantify the implications of the complications of periodontal pathology and dental mobility on the pathology of dysfunctional algo syndrome, a clinical entity with profound implications for the patient’s quality of life.

The working hypothesis is based on the existence of correlative aspects between malocclusion and temporomandibular joint pathology as a general trajectory. We aim to verify, by quantifiable methods, the way dental mobility influences static and dynamic occlusion with the identification of consequences at the TMJ level.

## 2. Aim of the Study

Our study aims to identify the correlations between dental mobility as a complication of periodontal disease and changes in dental occlusion, with the integration of these aspects in the evaluation of changes in the temporomandibular joint. We propose a gradual analysis of the factors involved in the triggering of dental mobility, from simple to complex and from the basal elements to the complex ones, with major implications in the dento- stomato-facial balance.

The elaboration of a precise diagnosis, based on the existing correlations between periodontal complications and the impairment of the parameters that characterize the morphology and functionality of the temporomandibular joint (TMJ), constitutes a relevant starting point for a complex integrative therapy at the level of the entire stomatognathic system.

Our findings emphasize the great importance of disease prevention and oral/periodontal healthcare for general well-being during a lifetime.

## 3. Materials and Methods

Clinical and laboratory evaluation was conducted in the 2018–2022 period, on a group of 110 women and 130 men, aged between 20–69, selected from our practice venue, Policlinica Stomatologica nr. 1 Iasi, Clinical Base of Dentistry Education “Mihail Kogalniceanu” Iasi, “Grigore T. Popa” University of Medicine and Pharmacy Iasi and “Apollonia” University Iasi. Overall, 125 subjects (57 women and 68 men) were diagnosed with periodontal disease with complications and TMJ disorders and followed periodontal therapy in the context of oral complex rehabilitation treatments (study group); the results of their clinical evaluation were compared with the results of the evaluation of the control group, made from the other 115 patients (53 women and 62 men). The control patients were diagnosed with incipient periodontal disease but with symptoms specific to the algo-dysfunctional syndrome.

The study design was built according to the methodology of case-control studies, where the selection of patients was consecutive for those presented in person for oral rehabilitation therapy. Patients presenting to the two medical units were selected based on the existing pathology, anchored in both the periodontal pathology registry and temporomandibular joint dysfunction, considering the inclusion and exclusion criteria, following the signing of the informed consent and the approval of the study protocol, approved by the institutional ethics committee (Protocol Code No. 547.10012012 and No. 176/17.04.2022).

The *inclusion criteria* for the cases were: patients suffering from periodontal disease, in different degrees, with, at the same time, the presence of symptoms specific to temporomandibular joint pathology.

The *exclusion criteria* for the cases were: non-periodontal disease, non-cooperating patients; subjects with advanced or terminal illnesses; oncological post-radiotherapy with poor blood coagulation degrees; incapacitated for scheduled procedures, people under medication that could possibly induce periodontological damages. Periodontal health may deteriorate more rapidly in diabetic patients with poorly controlled blood glucose levels than in other patients and they may not respond as well to traditional therapy. Additionally, periodontal alteration in rheumatoid arthritis and chronic renal disease are noted based on chronic inflammatory characteristics.

In accordance with the suggested methodology, we carried out an intraoral and extraoral clinical examination, evaluated the centric and posture relationships of the patients, palpated and percussioned the right and left TMJ joints, and assessed the muscular tonicity. We also conducted out an intraoral examination using static and dynamic occlusion parameter

The extra-oral examination was aimed at assessing the equality of the facial levels, a frequent change in occlusal disorders, as well as obvious changes at the articular level, and the intra-oral examination was based on the clinical procedures to record dental mobility: a pathological mobility inspection, a palpatory test, a percussion test, a dental pressure test, an automatic measurement using the Periotest (device produced by Siemens AG) and the measurement of gingival recession using periodontal probes of the CPITN. The dental mobility grades were assessed according to the Miller Classification. Observation recordings were completed for all patients including their personal data and their medical and oral history. As complementary examinations, we performed: a photographic examination and panoramic radiographs. Furthermore, CBCT. FOV 8.5 × 8.5 CBCTs were used to describe the detailed morphological bone descriptions with the precise identification of periodontal defects in patients examined with quantifiable occlusal changes and TMJ CBCTs (FOV 8 × 5) were used to evaluate the changes in the mandibular condyle and glenoid cavity, which were the basis for the development of an accurate diagnosis of joint dysfunction. The final result was obtained by multiplying all of the bleeding surfaces and dividing them by the number of dental surfaces examined, multiplying everything ×100.

Probing: the periodontal probe was used to measure the distance from the free marginal gum to the bottom of the pouch (sulcus) on the four sides of each tooth.

The result of the measures taken in the initial phase of the periodontal therapy was followed by evaluating two parameters: the plaque index (a faithful indicator of how patients were motivated and aware) and the PBI index (Papillary bleeding index) to assess gingival inflammation. These indices were recorded both at the beginning of treatment and during treatment sessions at six months.

### Statistical Analysis

The statistical analysis was performed in SPSS 27.0. The numerical variables were expressed through average values and standard deviations, and the categorial values were expressed through absolute frequencies and percentages. The Pearson Chi-squared test was used to compare the category variables within the samples. The value *p* < 0.05 was considered statistically significant and the value *p* < 0.01 was considered highly statistically significant.

The sample size of 240 patients was established according to the size of the population in Iasi county (around 850,000 inhabitants) and the prevalence of periodontal disease in Romania, which is around 45%, for a confidence level of 95% and a population proportion of 50% (standard value).

## 4. Results

The demographic characteristics of the analyzed samples are presented in Table 1. The sample structure in terms of genders and age groups is similar, the male patients being prevalent, as well as patients over 40 years old. Dental mobility and gingival recession were identified as more frequent in the study sample compared with the control sample, the differences being statistically significant in both cases. In total, 26.7% of patients had different types of TMJ disorders and 22.9% of patients had occlusal changes; the percentages are slightly increased in the study sample compared with the control sample, but the recorded differences are not statistically significant.

We evaluated the degree of dental mobility, as well as the degree of loss of periodontal attachment (gingival recession, depth of periodontal pockets) in the two samples and we found a certain association between their presence (58.7% of patients with gingival recession also have dental mobility, compared with only 17.6% of patients without gingival recession, *p* < 0.001).

We also evaluated the dental mobility and gingival recession presence on genders and age groups, by comparison in the study and the control samples. The results are presented in Table 2 and Table 3.

The statistical analysis identifies significant differences between the dental mobility presence in age groups, only in the study sample, and between the dental mobility degree in age groups, in the study sample as well as at the level of the global sample. We did not find significant differences between genders in which concerns the dental mobility presence or degree. We also noticed a prevalence of grade 2 dental mobility at age ranges between 50–59 years and 60–69 years, a clinical appearance that correlates with the presence of favorable factors represented by sublingual tartar factor, tooth mobility in quantifiable percentage (Table 2).

There are no significant differences between the gender or age groups which concern the presence of gingival recession; nevertheless, in the study sample, we noticed high percentages of gingival recession in young ages (66.7%—30–39 years), but this fact may be caused by other clinical conditions of the patients enrolled in our study (Table 3).

Regarding the degree of gingival recession, either directly or not correlated with dental mobility, it varies between 1–5 mm, and we identified statistically significant differences only between age groups at the level of the study sample and in the global sample. The gingival recession of 5 mm was found on a small number of teeth, in patients with ages between 50–69 years; the 1 mm recession was the most frequent in the age group 20–29 years (Table 3). The corroboration of the periodontal recession with the other specific elements of periodontal damage is a priority element in the specific therapeutic approach anchored in the reconstructive or specific register periodontal surgery, or, in the immobilization territory by means of periodontal immobilization systems followed by prosthetic restorations.

In order to gain the pathognomonic value, the dental mobility and the gingival recession must be correlated with the other signs present in the patient, particularly with the balance in the temporomandibular joint and occlusal changes.

We identified TMJ disorders in 64 patients (38 in the study sample and 26 in the control sample); the presence of TMJ disorders is statistically significantly associated with the presence of dental mobility and gingival recession at the level of both samples. The percentage of patients with TMJ disorders is significantly higher among patients with dental mobility (53.8%) compared with the patients without dental mobility (13.1%) in the whole sample, as well as separately in the study and the control samples (Table 4).

In addition to the periodontal disorders identified and quantified in the study group, 26.3% of the patients had temporomandibular joint damage on the left side, 18.4% of the patients presented impaired mandibular muscles on the left side, and 21.1% of the patients had damage to the centric relationship. Right-sided muscle damage was found also in 21.1% of patients, and postural relationship damage was observed in 13.2% of cases. In the control group, 26.9% of the left temporomandibular joint was affected, 11.5% of the left side muscles were affected and 19.2% of the centric relationship was impaired. As for the right-side manducatory muscles, it was affected in 19.2% of cases, and the postural relationship was affected in 23.1% of cases in the control group. These differences are not high enough to become statistically significant (Table 5, Figure 1).

Due to its infectious and degenerative pathology, periodontitis is an important etiological factor in triggering craniomandibular mal-relation, tooth mobility, tooth migration, and periodontal pain, shifting the way of receiving masticatory forces and influencing the mandibular dynamics to a significant extent. The impairment of the central relationship is a natural consequence of occlusal changes, due to the complications of dental mobility that are reflected in the aspects that are characteristic of TMJ algal dysfunction syndrome.

Periodontal damage may be the cause of pathological impairment, but it may also be the result of disorders of other elements of the stomatognathic system.

There are a number of periodontal changes that can have consequences that can trigger craniomandibular disorders. Malrelation may also have signs and symptoms that resonate periodically.

We identified occlusal changes in 55 patients (30 in the study sample and 25 in the control sample); their presence is statistically significantly associated with the presence of dental mobility and gingival recession, again at the level of both samples. The percentage of patients with occlusal changes is significantly higher among patients with dental mobility (43.8%) compared with the patients without dental mobility (12.5%) in the whole sample, as well as separately in the study and the control samples (Table 6).

In the patients with periodontal damage in the study group, the occlusal areas were affected in 16.7% of cases, compared with 12.0% of cases recorded in the control group; the supporting cusps were altered in 23.3% of cases in the study group and 20.0% of cases in the control group; the guiding cusps in the study group were affected in 26.7% of cases, while in the control group, the corresponding percentage is 24.0%. The Spee’s curve was damaged in 13.3% of cases in the study group and 20.0% of cases in the control group, and the cross-curve damage was noticed in 20.0% in the study group and in 24.0% in the control group. Again, even if there are some differences between the two studied samples, they are not statistically significant (Table 7, Figure 2).

The changes found during intraoral and extraoral examination are correlated with clinical signs of the dysfunctional syndrome of the stomatognathic system in different degrees and depending on an individual clinical case. For the patients of the study group, prophylactic and curative treatment was performed, including relaxation, periodontal etiological treatments and guided tissue regenerations for severe cases, as well as dental and endodontic treatments in order to restore the occlusal balance, guaranteeing future stabilizing prosthetic therapy. The prosthetic treatments were more inscribed in plural prosthetic works in the frontal area in a significant percentage, as well as in the lateral area. In some places, there were also five composite unit crowns.

There is a prevalence of periodontal, endodontic and edentulous impairments, as well as TMJ dysfunctions, correlated with the changes induced by dental mobility within the occlusal parameters.

Regarding the need for rehabilitation therapies in the study and the control groups, the need for craniomandibular repositioning was 13.6% in the study group and 8.7% in the control group, the need for conditioning braces was 15.2% in the study group and 8.7% in the control group, while the requirement for periodontal etiological therapy was 16.8% in the study group and 27.8% in the control group. In terms of periodontal surgery, the intervention was 8.8% in the study group and 7.0% in the control group, while the need for dental therapy was also 8.8% in the study group, but 10.4% in the control. Through the quantification of the endodontic therapy requirement, we recorded a 13.6% need in the study group and 15.7% in the control group—the same percentages were also observed for the need for fixed front-lateral prostheses. The need for a fixed prosthesis in the lateral area slightly prevailed in the study group, in a percentage of 4.8% compared to 3.5% in the control group, while unidentate prosthesis prevailed in a percentage of 4.8% in the study group compared to 2.6% in the control group (Table 8, Figure 3).

## 5. Discussions

It will not always be possible to achieve a total reversal of destructive tendencies occurring in the dental system; however, ceasing them will lead to the maintenance of systemic homeostasis. It is a mistake to believe that dental disease is due to one cause alone; rather, it is usually the result of the action of multiple factors that can be divided into non-specific and specific determinants and promoters. The body’s response is a function of its resistance to the intensity and duration of the aggression. The removal of the triggering factor (well identified) will lead to the restoration of health. In addition, a determining factor, even if it acts on only one systemic element, will lead to disturbances in the entire stomatognathic system. As a result, an occlusal interference due to iatrogenic over occlusion may initially cause dental sensitivity to percussion, dental mobility and abrasion in an attempt to compensate for the imbalance, with subsequent changes in the mandibular dynamic trajectories, to avoid the obstacle [15,16]. The complexity and individuality of each clinical case, the multitude of the clinical phenomena and correlations that are established between the elements of the dental system, and the avalanche of modern technical means of investigation are all particular situations that require an increased level of competence and performance in the diagnostic and therapeutic decision [17,18].

The diagnosis of PD is achieved by inspecting the adjacent gum teeth tissue through the quantification of specific indexes, both visually and with a probe, in addition to X-rays looking for bone loss damages. The World Health Organization (WHO) has maintained a global oral health data bank using the community periodontal index (CPI) [19,20].

In recent decades, the means of diagnosis have intensified, and new treatment schemes have been designed, but all start from simple means, as the first stage of etiological therapy [21,22]

Carranza and Newman consider the introduction of etiological therapy as the first phase of periodontal therapy to be of paramount importance. In fact, in the prognosis of periodontal disease, the degree of the cooperation of the patient with the doctor, in terms of patient awareness, and the establishment of rigorous measures of oral hygiene are taken into account. They believe that this therapy intercepts periodontal disease by eliminating and controlling all of the etiological factors involved [23,24].

All of these disorders will result in the request of other dental periodontal units, muscular hyperactivity, trismus, headache, stress, depression, functional and then morphological disorders of the meniscus-condylar ensemble, arthrosis of the temporomandibular joint, etc. This example seeks to highlight the multitude of signs and symptoms that are caused by a single, seemingly trivial factor. Once eliminated, the factor before the elements of the stomatognathic system change, becoming irreversible, the symptoms disappear, and the systemic resistance is not affected.

In the range of causes that can lead to major disturbances of the stomatognathic system, dental mobility occupies a majority percentage, a clinical aspect frequently found in periodontal pathology and occlusal trauma.

Normal, physiologic tooth mobility of approximately 0.25 mm is currently found in healthy patients. This is because the tooth is not fused to the bones of the jaws, but is connected to the sockets by the periodontal ligament. This slight mobility is to accommodate forces on the teeth during chewing without causing damage. Deciduous teeth also naturally become looser just before their exfoliation. This is caused by the gradual resorption of their roots, stimulated by the developing permanent tooth underneath.

Occlusal trauma occurs when excessive force is set on teeth. With periodontal disease, there can be irreversible trauma to the teeth [25,26].

According to the SDCEP guidelines, when teeth have either over-erupted or drifted due to periodontal disease, it is recommended to check for fremitus or occlusal interference (Figure 4).

When a tooth occludes in an undesirable contact point, it prevents the other teeth from achieving the ideal and harmonious contact points. There are four types of occlusal interference: *centric*, *working*, *non-working and protrusive*.

Occlusal interference can be managed by removing the premature contact point or through restorative materials.

Rehabilitation treatment through prophylaxis or curative principles, as well as medicinal to surgical treatment accompanied by prosthetic treatments, can be viewed with a positive performance in terms of increasing the functionality of the stomatognathic system.

There is now good evidence that stress on the temporomandibular joints due to the absence of teeth in the posterior areas of the dental arches accentuates the signs and symptoms of dysfunction, but the causal link is not certain, in the sense that either dental occlusion causes the dysfunction or it results from the dysfunctional condition. The most recent evidence in the literature suggests that dental occlusion can be attributed to a secondary role in the onset of craniomandibular dysfunction, following trauma, oral parafunction, stress or dental iatrogenesis (Figure 5).

The following are warning signs of periodontal disease: red or swollen gums, tender or bleeding gums, painful chewing, loose teeth, sensitive teeth, retracted gums and changes in the way the teeth are positioned [27,28].

Certain factors increase the risk of periodontal disease: stress, smoking, heredity diabetes mellitus, poor oral hygiene, crooked teeth, underlying mmune-deficiencies, such as AIDS, fillings that have become defective, medications that cause dry mouth, bridges that no longer fit properly and female hormonal changes, such as with pregnancy or the use of oral contraceptives [29,30]. The participants’ gingival status was assessed using the WHO periodontal probe, measuring four gingival units (mesial, distal, vestibular, lingual) corresponding to each of those present in the oral cavity and observing whether or not bleeding was present (Figure 6).

The study found statistically significant correlations between dental mobility and the degree of vertical resorption quantified on CBCT, with relevance to changes at the articular level and reflected by changes in shape or parameter changes during dynamic occlusion. It should also be noted that although the vertical lysis shows significant dimensions, the changes at the articular level captured on CBCT are not significant, which can be explained by the myocene influences at this level (Figure 7).

Regarding the paraclinical grounds, several sections detected on the CBCT, offering horizontal and vertical selective images with precision, are quite significant. The combination of the two observed resorptions is the basis for the clinical modifications induced by dental mobility. Thus, 25% of the study group followed the specialized imaging evaluations by bringing TMJ tomographs in the paraclinical register and the CBCT of TMJ.

In the treatment of the complications of dental mobility of the craniomandibular mal-relation and of other diseases of the stomatognathic system, it is necessary to apply a therapy that holds a special place as an approach and result, namely occlusal balancing. A precise occlusal rebalancing is the desideratum of all dentists and can be assimilated into the generic term of occlusal therapy. Occlusal therapy has broad indications, but the etiological factors of occlusal issues must be established from the beginning, with chaotic organization being accompanied by failure [31,32].

Occlusal balancing refers to the correction of non-harmonic occlusal contacts and the remodeling of natural and artificial dental surfaces that interfere during the functional cycles of the stomatognathic system: unblocking the mandibular movements, transmitting the occlusion forces correctly, the uniform distribution of the inter-arcade contact points in centric relation and maximum intercuspidation, restoring the morphological balance of the teeth, obtaining a dynamic occlusion, terminal occlusions, optimal joint and muscular functionalities [33,34].

Occlusal balancing treatment can be performed through several methods, with different degrees of preservation applied depending on the form of the craniomandibular mal-relation [35,36].

Interferences on the inactive side due to dental mobility are considered harmful because the pressure exerted on this side is increased by the fact that it is close to the condyles, and on the other hand, it is directed towards inclined slopes, decomposing into overloading horizontal forces. In addition, non-working interferences, depending on the direction of the force, tend to cause torsions or rotations. The use of the occlusion stability criteria simplifies the processes of diagnosis and drawing up an optimal treatment plan [37].

According to De Boever et al. (2008), there are two mechanisms that could incriminate the occlusal factor as an etiological one for the onset of craniomandibular dysfunction: an acute change in occlusion and significant orthopedic instability, with the over-straining of the craniomandibular musculoskeletal structures [38].

Extensive studies over the last two decades have concluded that the relationship between the well-documented occlusal factors and TMJ dysfunction is insufficiently verified. Studies by C. Sadowski et al. also found no associations between the functional occlusal relationships, such as the centric-maximal intercuspidation relationship, orthodontic abnormalities or joint noises [39].

Regarding the correlative aspects between dental mobility, occlusal dysfunction and temporomandibular joint implications, we noted particular aspects, such as sudden malocclusion due to muscle dysfunction, with various degrees of importance depending on the muscle involved. In the situation of spasm at the level of the inferior fasciculus of the external pterygoid, the mandibular condyle on that side is displaced slightly forward into the glenoid cavity. This will result in the inoculation of the ipsilateral posterior teeth and excessive contralateral tooth contact in the canine region. If the muscle spasm is confined to the mandibular levator musculature, the patient will suddenly notice that the teeth are no longer touching evenly, a change that is hardly noticeable objectively.

The most common therapy during joint dysfunction is the use of a splint, with important aspects in the therapeutic plan being related to the periodontal status, i.e., the presence of dental mobility. Before choosing the type of device, the patient should know that there are possible negative consequences of this therapy, manifested by irreversible alterations of occlusion (e.g., posterior inocclusion). The stabilization devices mentioned above can achieve many of the objectives pursued by the repositioning prosthesis and this is with less risk. The repositioning device is made on the dental arch, covering the occlusal surface of all the teeth, with indentations or guide ramps that can produce the advancement of the mandible in a less painful and therapeutic meniscus-condyle relationship, a position located just before the cracks occur in the closing arch of the mandible. The repositioning device, as opposed to a relaxing brace, is more effective if worn permanently [40,41].

The choice of the type of occlusal therapy is very important and is often a difficult task facing the practitioner. In the vast majority of cases, the choice has to be made between selective grinding, the use of unidentate or plural fixed prostheses and orthodontic therapy. Often, the critical factor determining the appropriate treatment protocol is the interarch discrepancy in the buccal-oral direction, assessed between the maxillary and mandibular posterior teeth. This relationship should be examined after placing the mandibular condyles in the orthopedically stable musculoskeletal position (centric relationship). In this situation, the mandible is gently guided into intercuspidation by rotations in the axis of the terminal hinge, following the moment when the first interdental contact occurs. Before choosing the irreversible occlusal therapeutic management, it is absolutely necessary to mount the study and diagnostic models in a simulator of mandibular kinematics based on the facebow recording. It should not be forgotten that there is currently no scientific-based evidence that selective preventive or therapeutic grinding is effective in this area, so it is considered that occlusal adjustment cannot be recommended for the prevention or treatment of craniomandibular dysfunction [42,43].

The implications of occlusal changes at the level of the temporomandibular joint in patients with periodontal disease are explained by the known fact that dental occlusion is the key to all dentistry, having the role of stabilizing the mandible in its positions relative to the skull and participating in the achievement of systemic functions. The occlusal surfaces of the teeth participate in mutual protection, playing an important role in ensuring optimal periodontal, muscular and articular functionality. Any distortion of the occlusal plane or any occlusal interference can trigger pathological changes in the elements of the stomatognathic system [44].

Occlusal disorders cannot act alone to alter the mandible-cranial relationships but through the masticatory muscles and the two temporomandibular joints.

Occlusal disorders detected in patients with advanced periodontal disease resulted in alterations in the static and dynamic mandible-cranial relationships, subsequently entraining and dysfunctioning other systemic elements.

Occlusal disorders occurred as a consequence of vertical dental migrations associated with dental mobility. Consequently, occlusal dysfunctions result from the action of any factor that alters the balance; harmony between the two dento-alveolar arches is achieved by means of occlusal curves, occlusal plane and mandibular dynamic patterns with dental contact [45]. Clinically, it manifests itself in the form of premature contacts, occlusal interferences or localized or generalized dental abrasion

Tooth mobility resulting in a shift in the occlusal areas, changes in the integrity and shape of the supporting and guiding cusps, altered occlusal curves (interrupted, inverted, horizontal) and inverted, uneven occlusal plane is an important factor of occlusal dysfunction, resulting in changes in the mandibular dynamics patterns, given the role of dental determinants in the realization of the mandibular dynamics.

The early detection of temporomandibular joint-specific changes as a complication of occlusal changes related to dental mobility, an important complication of periodontal disease, is a particularly important desideratum that creates the prerequisites for optimal quality of life for patients, as it is known that the symptoms of joint dysfunction disturb the functionality of the dental-maxillary apparatus, even taking on a disabling character.

The limitations of this study are due to the fact that the CBCT of the temporomandibular joint could not be performed on all of the subjects in the study group, but only on those with occlusal changes correlated with tooth mobility and the presence of symptoms specific to temporomandibular joint dysfunctional syndromes.

Unilateral mastication, frequently encountered in the presence of dental mobility in a particular group of teeth, has led to unequal right/left muscle contractions, asymmetric articular stress, asymmetrizing the mandibular dynamics trajectories with and without dental contact and, thus, implicitly impacting the craniomandibular dynamic relationships. The future directions of study aim to extend the correlative aspects between the periodontal damage and the general condition of the patient and the existing morphological and functional changes in the temporomandibular joint territory. We aim to carry out a prognosis of temporomandibular pathology in a systemic context with different evolutionary trajectories of periodontal damage by means of a digital application.

The risk of developing craniomandibular dysfunction is higher when several predisposing factors are associated, with periodontal complications playing an important role. Through its infectious and degenerative pathology, the periodontium represents an important etiological factor in triggering craniomandibular mal-relation, tooth mobility, dental migrations and periodontal pain, altering the way masticatory forces are received and influencing the mandibular dynamics by 90%, finally closing a dysfunctional vicious circle.

## 6. Conclusions

The evaluation of the prevalence and degree of the loss of periodontal attachment (gingival recession, depth of periodontal pockets) demonstrates that there is a direct relationship between dental mobility and changes at the occlusal level at the level of the temporomandibular joint.

The correlation of the specific signs of temporomandibular pathology in 64 patients correlated with the presence of tooth mobility and gingival recession in both samples. This is an eloquent plea for the early initiation of therapy in the temporomandibular registry in patients with periodontal disease.

In the patients with periodontal damage in the study group, the occlusal areas were affected in 16.7% of cases, compared with 12.0% of cases recorded in the control group; the supporting cusps were altered in 23.3% of cases in the study group and 20.0% of cases in the control group; the guiding cusps in the study group were affected in 26.7% of cases, while in the control group, the corresponding percentage is 24.0%.

The importance of clinical and paraclinical CBCT assessments in the elective diagnosis of periodontal pathology and its collateral effects at the level of the temporomandibular joint plays a defining role in the development of interdisciplinary therapy with good long-term results.

## Figures and Tables

**Figure 1 diagnostics-13-00598-f001:**
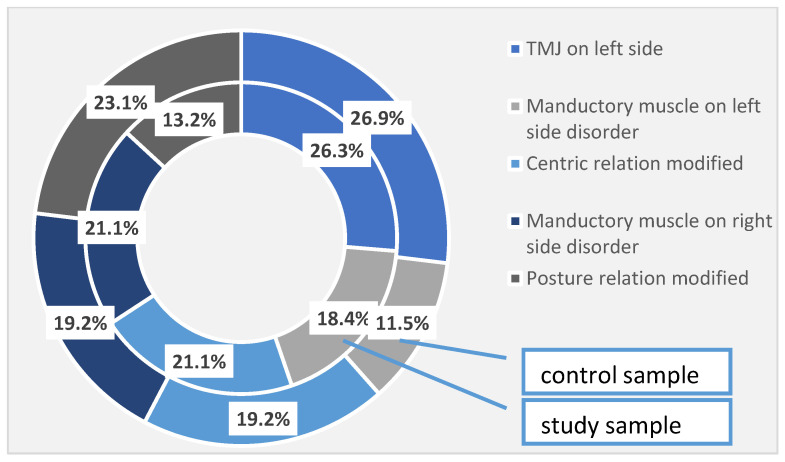
TMJ disorders identified at the investigated patients (the total sample).

**Figure 2 diagnostics-13-00598-f002:**
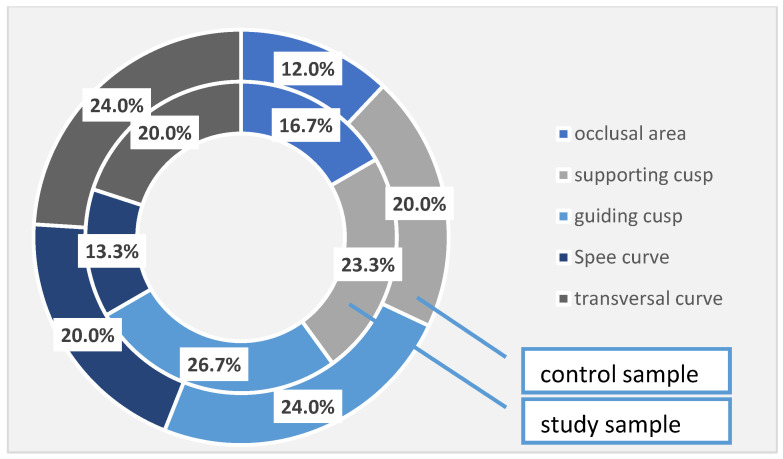
Occlusal changes identified at the investigated patients (the total sample).

**Figure 3 diagnostics-13-00598-f003:**
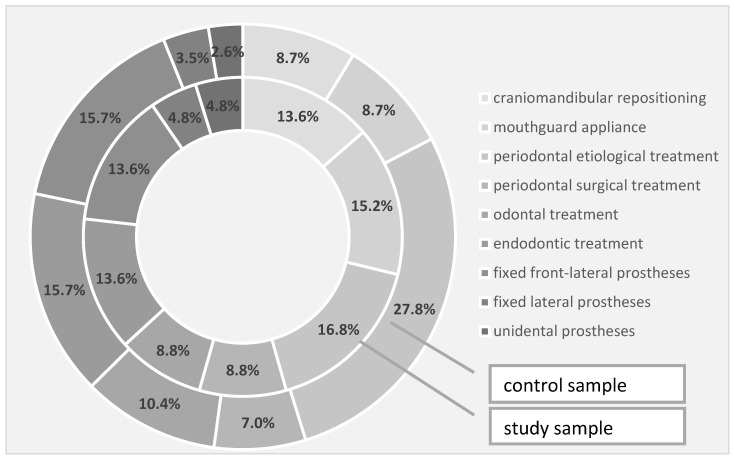
Treatment requirements identified in the investigated samples.

**Figure 4 diagnostics-13-00598-f004:**
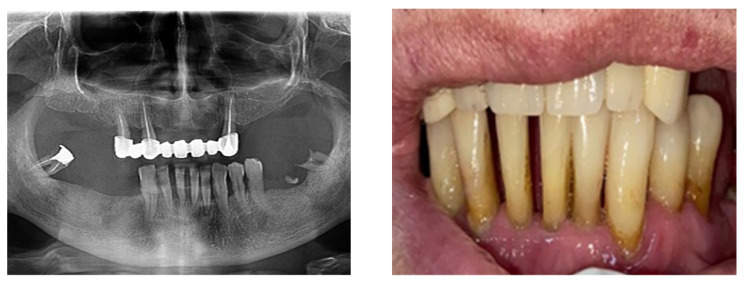
Clinical and paraclinical aspects of dental mobility. (**a**) Paraclinical aspects of dental mobility; (**b**) Clinical aspects of dental mobility.

**Figure 5 diagnostics-13-00598-f005:**
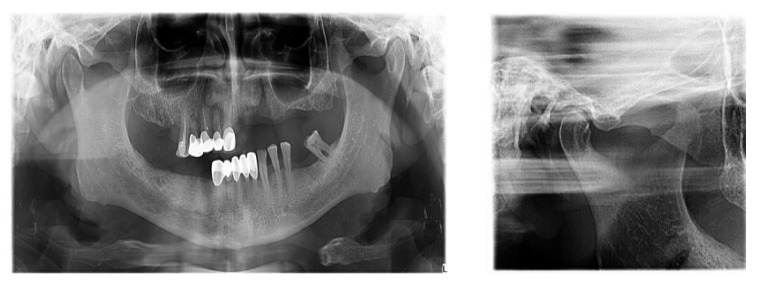
TMJ tomography aspects in patients with periodontal injuries, open and shut position. (**a**) OPT with periodontal injuries; (**b**) TMJ aspect.

**Figure 6 diagnostics-13-00598-f006:**
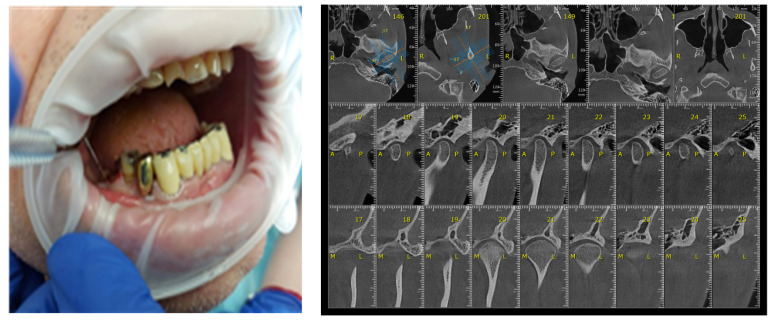
Correlation between Periodontal probing and TMJ exam. (**a**) Clinical aspects of periodontal damage; (**b**) TMJ changes related to periodontal status.

**Figure 7 diagnostics-13-00598-f007:**
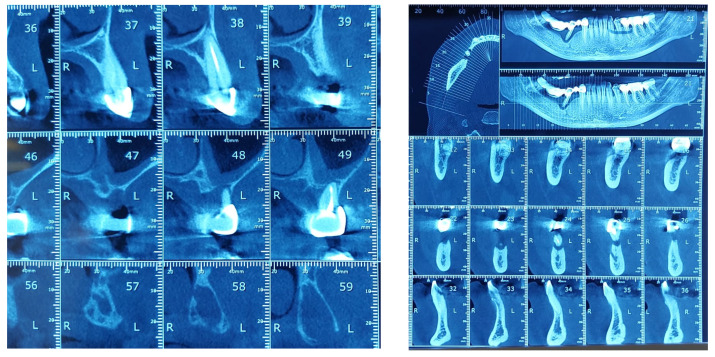
Bone Changes in periodontal disease through CBCT. (**a**) Maxilla CBCT- Bone Changes in periodontal disease; (**b**) CBCT mandible- Bone Changes in periodontal disease.

**Table 1 diagnostics-13-00598-t001:** Demographic and clinical parameters of the sample.

Parameter	Total (*n* = 240)		Study Sample (*n* = 125)		Control Sample (*n* = 115)		*p*-Value
	*n*	%	*n*	%	*n*	%	
Gender							0.940
M	130	54.2%	68	54.4%	62	53.9%	
F	110	45.8%	57	45.6%	53	46.1%	
Age group							0.498
20–29 ys	30	12.5%	16	12.8%	14	12.2%	
30–39 ys	32	13.3%	12	9.6%	20	17.4%	
40–49 ys	52	21.7%	28	22.4%	24	20.9%	
50–59 ys	68	28.3%	36	28.8%	32	27.8%	
60–69 ys	58	24.2%	33	26.4%	25	21.7%	
Dental mobility							0.011 *†
absent	160	66.7%	74	59.2%	86	74.8%	
present	80	33.3%	51	40.8%	29	25.2%	
Gingival recession							0.007 **†
absent	148	61.7%	67	53.6%	81	70.4%	
present	92	38.3%	58	46.4%	34	29.6%	
TMJ disorders							0.173
absent	176	73.3%	87	69.6%	89	77.4%	
present	64	26.7%	38	30.4%	26	22.6%	
Occlusal changes							0.677
absent	185	77.1%	95	76.0%	90	78.3%	
present	55	22.9%	30	24.0%	25	21.7%	

† Pearson Chi-squared test; * *p* < 0.05 statistically significant; ** *p* < 0.01 statistically highly significant.

**Table 2 diagnostics-13-00598-t002:** a. The dental mobility presence by genders and age groups in the investigated samples. b. The dental mobility degree by genders and age groups in the investigated samples.

	**a**											
	**Dental Mobility**											
**Parameter**	**Total ** **(*n* = 240)**				**Study Sample ** **(*n* = 125)**				**Control Sample** **(*n* = 115)**			
	**Absent** ***n* (%)**		**Present** ***n* (%)**		**Absent** ***n* (%)**		**Present** ***n* (%)**		**Absent** ***n* (%)**		**Present** ***n* (%)**	
Gender	*p* = 0.119 †				*p* = 0.120 †				*p* = 0.556 †			
M	81 (62.3%)		49 (37.7%)		36 (52.9%)		32 (47.1%)		45 (72.6%)		17 (27.4%)	
F	79 (71.8%)		31 (28.2%)		38 (66.7%)		19 (33.3%)		41 (77.4%)		12 (22.6%)	
Age group	*p* = 0.379 †				*p* = 0.038 *†				*p* = 0.503 †			
20–29 ys	22 (73.3%)		8 (26.7%)		10 (62.5%)		6 (37.5%)		12 (85.7%)		2 (14.3%)	
30–39 ys	17 (53.1%)		15 (46.9%)		2 (16.7%)		10 (83.3%)		15 (75.0%)		5 (25.0%)	
40–49 ys	34 (65.4%)		18 (34.6%)		17 (60.7%)		11 (39.3%)		17 (70.8%)		7 (29.2%)	
50–59 ys	49 (72.1%)		19 (27.9%)		23 (63.9%)		13 (36.1%)		26 (81.3%)		6 (18.8%)	
60–69 ys	38 (65.5%)		20 (34.5%)		22 (66.7%)		11 (33.3%)		16 (64.0%)		9 (36.0%)	
	**b**											
	**Dental Mobility Degree**											
**Parameter**	**Total ** **(*n* = 80)**				**Study Sample ** **(*n* = 29)**				**Control Sample** **(*n* = 51)**			
	**I** ***n* (%)**	**II** ***n* (%)**		**III** ***n* (%)**	**I** ***n* (%)**	**II** ***n* (%)**		**III** ***n* (%)**	**I** ***n* (%)**	**II** ***n* (%)**		**III** ***n* (%)**
Gender	*p* = 0.413 †				*p* = 0.378 †				*p* = 0.563 †			
M	13 (26.5%)	22 (44.9%)		14 (28.6%)	11 (34.4%)	12 (37.5%)		9 (28.1%)	2 (11.8%)	10 (58.8%)		5 (29.4%)
F	8 (25.8%)	10 (32.3%)		13 (41.9%)	5 (26.3%)	5 (26.3%)		9 (47.4%)	3 (25.0%)	5 (41.7%)		4 (33.3%)
Age group	*p* = 0.001 **†				*p* = 0.004 **†				*p* = 0.257 †			
20–29 ys	6 (75.0%)	2 (25.0%)		-	5 (83.3%)	1 (16.7%)		-	1 (50.0%)	1 (50.0%)		-
30–39 ys	5 (33.3%)	7 (46.7%)		3 (20.0%)	5 (50.0%)	3 (30.0%)		2 (20.0%)	-	4 (80.0%)		1 (20.0%)
40–49 ys	8 (44.4%)	4 (22.2%)		6 (33.3%)	5 (45.5%)	3 (27.3%)		3 (27.3%)	3 (42.9%)	1 (14.3%)		3 (42.9%)
50–59 ys	1 (5.3%)	11 (57.9%)		7 (36.8%)	1 (7.7%)	7 (53.8%)		5 (38.5%)	-	4 (66.7%)		2 (33.3%)
60–69 ys	1 (5.0%)	8 (40.0%)		11 (55.0%)	-	3 (27.3%)		8 (72.7%)	1 (11.1%)	5 (55.6%)		3 (33.3%)

† Pearson Chi-squared test; * *p* < 0.05 statistically significant; ** *p* < 0.01 statistically highly significant.

**Table 3 diagnostics-13-00598-t003:** a. The gingival recession presence by genders and age groups in the investigated samples. b. The gingival recession degree by genders and age groups in the investigated samples.

	**a**									
	**Gingival Recession**									
**Parameter**	**Total ** **(*n* = 240)**			**Study Sample ** **(*n* = 125)**				**Control Sample** **(*n* = 115)**		
	**Absent** ***n* (%)**	**Present** ***n* (%)**		**Absent** ***n* (%)**		**Present** ***n* (%)**		**Absent** ***n* (%)**	**Present** ***n* (%)**	
Gender	*p* = 0.450 †			*p* = 0.594 †				*p* = 0.683 †		
M	83 (63.8%)	47 (36.2%)		38 (55.9%)		30 (44.1%)		45 (72.6%)	17 (27.4%)	
F	65 (59.1%)	45 (40.9%)		29 (50.9%)		28 (49.1%)		36 (67.9%)	17 (32.1%)	
Age group	*p* = 0.677 †			*p* = 0.333 †				*p* = 0.703 †		
20–29 ys	20 (66.7%)	10 (33.3%)		9 (56.3%)		7 (43.8%)		11 (78.6%)	3 (21.4%)	
30–39 ys	20 (62.5%)	12 (37.5%)		4 (33.3%)		8 (66.7%)		16 (80.0%)	4 (20.0%)	
40–49 ys	33 (63.5%)	19 (36.5%)		16 (57.1%)		12 (42.9%)		17 (70.8%)	7 (29.2%)	
50–59 ys	44 (64.7%)	24 (35.3%)		23 (63.9%)		13 (36.1%)		21 (65.6%)	11 (34.4%)	
60–69 ys	31 (53.4%)	27 (46.6%)		15 (45.5%)		18 (54.5%)		16 (64.0%)	9 (36.0%)	
	**b**									
	**Gingival Recession Degree**									
	**1 mm** ***n* (%)**		**2 mm** ***n* (%)**		**3 mm** ***n* (%)**		**4 mm** ***n* (%)**			**5 mm** ***n* (%)**
**Total (*n* = 92)**										
Gender	*p* = 0.149 †									
M	15 (31.9%)		16 (34.0%)		4 (8.5%)		7 (14.9%)			5 (10.6%)
F	13 (28.9%)		12 (26.7%)		13 (28.9%)		4 (8.9%)			3 (6.7%)
Age group	*p* = 0.000 **†									
20–29 ys	9 (90.0%)		1 (10.0%)		-		-			-
30–39 ys	5 (41.7%)		7 (58.3%)		-		-			-
40–49 ys	5 (26.3%)		8 (42.1%)		5 (26.3%)		1 (5.3%)			-
50–59 ys	5 (20.8%)		7 (29.2%)		6 (25.0%)		3 (12.5%)			3 (12.5%)
60–69 ys	4 (14.8%)		5 (18.5%)		6 (22.2%)		7 (25.9%)			5 (18.5%)
**Study sample (*n* = 58)**										
Gender	*p* = 0.285 †									
M	10 (33.3%)		10 (33.3%)		3 (10.0%)		4 (13.3%)			3 (10.0%)
F	5 (17.9%)		8 (28.6%)		9 (32.1%)		4 (14.3%)			2 (7.1%)
Age group	*p* = 0.000 **†									
20–29 ys	7 (100.0%)		-		-		-			-
30–39 ys	3 (37.5%)		5 (62.5%)		-		-			-
40–49 ys	2 (16.7%)		6 (50.0%)		4 (33.3%)		-			-
50–59 ys	-		4 (30.8%)		4 (30.8%)		3 (23.1%)			2 (15.4%)
60–69 ys	3 (16.7%)		3 (16.7%)		4 (22.2%)		5 (27.8%)			3 (16.7%)
**Control sample (*n* = 34)**										
Gender	*p* = 0.183 †									
M	5 (29.4%)		6 (35.3%)		1 (5.9%)		3 (17.6%)			2 (11.8%)
F	8 (47.1%)		4 (23.5%)		4 (23.5%)		-			1 (5.9%)
Age group	*p* = 0.767 †									
20–29 ys	2 (66.7%)		1 (33.3%)		-		-			-
30–39 ys	2 (50.0%)		2 (50.0%)		-		-			-
40–49 ys	3 (42.9%)		2 (28.6%)		1 (14.3%)		1 (14.3%)			-
50–59 ys	5 (45.5%)		3 (27.3%)		2 (18.2%)		-			1 (9.1%)
60–69 ys	1 (11.1%)		2 (22.2%)		2 (22.2%)		2 (22.2%)			2 (22.2%)

† Pearson Chi-squared test; ** *p* < 0.01 statistically highly significant.

**Table 4 diagnostics-13-00598-t004:** The TMJ disorders presence by dental mobility and gingival recession in the investigated samples.

	TMJ Disorders					
Parameter	Total (*n* = 240)		Study Sample (*n* = 125)		Control Sample (*n* = 115)	
	Absent *n* (%)	Present *n* (%)	Absent *n* (%)	Present *n* (%)	Absent *n* (%)	Present *n* (%)
Dental mobility	*p* = 0.000 **†		*p* = 0.000 **†		*p* = 0.000 **†	
absent	139 (86.9%)	21 (13.1%)	62 (83.8%)	12 (16.2%)	77 (89.5%)	9 (10.5%)
present	37 (46.3%)	43 (53.8%)	25 (49.0%)	26 (51.0%)	12 (41.4%)	17 (58.6%)
Gingival recession	*p* = 0.000 **†		*p* = 0.001 **†		*p* = 0.002 **†	
absent	124 (83.8%)	24 (16.2%)	55 (82.1%)	12 (17.9%)	69 (85.2%)	12 (14.8%)
present	52 (56.5%)	40 (43.5%)	32 (55.2%)	26 (44.8%)	20 (58.8%)	14 (41.2%)

† Pearson Chi-squared test; ** *p* < 0.01 statistically highly significant.

**Table 5 diagnostics-13-00598-t005:** TMJ disorders identified in the investigated samples. † Pearson Chi-squared test.

	Total (*n* = 240)		Study Sample (*n* = 125)		Control Sample (*n* = 115)		*p*-Value
	*n*	%	*n*	%	*n*	%	
**TMJ disorder**							0.843 †
TMJ on left side	17	26.6%	10	26.3%	7	26.9%	
Manductory muscle on left side disorder	10	15.6%	7	18.4%	3	11.5%	
Centric relation modified	13	20.3%	8	21.1%	5	19.2%	
Manductory muscle on right side disorder	13	20.3%	8	21.1%	5	19.2%	
Posture relation modified	11	17.2%	5	13.2%	6	23.1%	
Total	64	100.0%	38	100.0%	26	100.0%	

**Table 6 diagnostics-13-00598-t006:** The occlusal changes presence by dental mobility and gingival recession in the investigated samples.

	Occlusal Changes					
Parameter	Total (*n* = 240)		Study Sample (*n* = 125)		Control Sample (*n* = 115)	
	Absent *n* (%)	Present *n* (%)	Absent *n* (%)	Present *n* (%)	Absent *n* (%)	Present *n* (%)
Dental mobility	*p* = 0.000 **†		*p* = 0.000 **†		*p* = 0.000 **†	
absent	140 (87.5%)	20 (12.5%)	66 (89.2%)	8 (10.8%)	74 (86.0%)	12 (14.0%)
present	45 (56.3%)	35 (43.8%)	29 (56.9%)	22 (43.1%)	16 (55.2%)	13 (44.8%)
Gingival recession	*p* = 0.000 **†		*p* = 0.000 **†		*p* = 0.005 **†	
absent	131 (88.5%)	17 (11.5%)	62 (92.5%)	5 (7.5%)	69 (85.2%)	12 (14.8%)
present	54 (58.7%)	38 (41.3%)	33 (56.9%)	25 (43.1%)	21 (61.8%)	13 (38.2%)

† Pearson Chi-squared test; ** *p* < 0.01 statistically highly significant.

**Table 7 diagnostics-13-00598-t007:** Occlusal changes identified in the investigated samples.

	Total (*n* = 240)		Study Sample (*n* = 125)		Control Sample (*n* = 115)		*p*-Value
	*n*	%	*n*	%	*n*	%	
**Occlusal changes**							0.941 †
occlusal area	8	14.5%	5	16.7%	3	12.0%	
supporting cusp	12	21.8%	7	23.3%	5	20.0%	
guiding cusp	14	25.5%	8	26.7%	6	24.0%	
Spee curves	9	16.4%	4	13.3%	5	20.0%	
transversal curves	12	21.8%	6	20.0%	6	24.0%	
Total	55	100.0%	30	100.0%	25	100.0%	

† Pearson Chi-squared test.

**Table 8 diagnostics-13-00598-t008:** Treatment requirements identified in the investigated samples. † Pearson Chi-squared test.

	Total (*n* = 240)		Study Sample (*n* = 125)		Control Sample (*n* = 115)		*p*-Value
	*n*	%	*n*	%	*n*	%	
**Treatment requirements**							0.390 †
craniomandibular repositioning	27	11.3%	17	13.6%	10	8.7%	
mouthguard appliance	29	12.1%	19	15.2%	10	8.7%	
periodontal etiological treatment	53	22.1%	21	16.8%	32	27.8%	
periodontal surgical treatment	19	7.9%	11	8.8%	8	7.0%	
odontal treatment	23	9.6%	11	8.8%	12	10.4%	
endodontic treatment	35	14.6%	17	13.6%	18	15.7%	
fixed front-lateral prostheses	35	14.6%	17	13.6%	18	15.7%	
fixed lateral prostheses	10	4.2%	6	4.8%	4	3.5%	
unidental prostheses	9	3.8%	6	4.8%	3	2.6%	

## Data Availability

Not applicable.

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
