# Peer review of "Research Regarding Dental Mobility Phenomena in the Clinical Recognition Diagnosis of Temporomandibular Disorders"

_diagnostics, 2023, doi:10.3390/diagnostics13040598_

Round 1

Reviewer 1 Report (Previous Reviewer 1)

Abstract is too long, please shorten it and limit it to the most important information about the study, in the abstract some sentences are written in a different font size. 

Please make the purpose of the study a little more specific.

On what basis was the study group selected? Were they consecutive extinguishers of patients?

According to what classification was the degree of periodontal disease determined? 

You write: "people under medication that could possibly induce periodontal damage" what does this mean? these diseases should be listed. 

You write: "In the context of the proposed methodology, we performed an extraoral and intraoral clinical examination".

These examinations should be described. 

Such as here: https://doi.org/10.3390/ijerph19138139

Which CBCT was performed - what FOV? What was evaluated 

How was tooth mobility studied- the methodology should be described, such as here:  https://doi.org/10.3390/jcm10122655

In the abstract and introduction you write about Quality of Life, but later in the manuscript it is no longer repeated. was QoL studied? if so describe it, with what form?

describe the limitations of the study

the conclusions are very long, shorten them - so that they correspond directly to the research question posed,

there are errors in the list of abbreviations, correct them

Bibliography could be more up-to-date

Author Response

Good afternoon,

           Thank you ,for your time in reviewing our study carefully, for publication, we have seriously corrected with track changes, the important things you have suggested to us:

Abstract is too long, please shorten it and limit it to the most important information about the study, in the abstract some sentences are written in a different font size. -Done

Please make the purpose of the study a little more specific.-done

On what basis was the study group selected? Were they consecutive extinguishers of patients?- modified

According to what classification was the degree of periodontal disease determined? -Miller classification  

You write: "people under medication that could possibly induce periodontal damage" what does this mean? these diseases should be listed.- Added

You write: "In the context of the proposed methodology, we performed an extraoral and intraoral clinical examination".

These examinations should be described. - Modified

Such as here: https://doi.org/10.3390/ijerph19138139 --added on references

Which CBCT was performed - what FOV? What was evaluated -done

How was tooth mobility studied- the methodology should be described, such as here:  https://doi.org/10.3390/jcm10122655 --added

In the abstract and introduction, you write about Quality of Life, but later in the manuscript it is no longer repeated. was QoL studied? if so describe it, and with what form?- Done

Describe the limitations of the study-   added

The conclusions are very long, shorten them - so that they correspond directly to the research question posed,  modified

There are errors in the list of abbreviations, correct them  -done

The bibliography could be more up-to-date- corrected and added.

Reviewer 2 Report (New Reviewer)

Dear Authors the paper  is really interesting, and fits the objectives of the journal; 

-About the Title of the article, I suggest you to modify it and add the type of article.

-Please be sure to use only keywords accordingly to medical subject headings (Mesh word) for a better indexing.

- The introduction section is very short and is needed to add other references to increase the quality of the manuscript, 

I suggest you some articles:  [DOI: 10.1080/08869634.2022.2137129] ; [DOI: 10.1155/2022/7091153] ;   [https://doi.org/10.3390/prosthesis4020025] ; [/doi/full/10.1080/08869634.2022.2126079]

-You need to review the grammar and English of your article, 

Figures are blurry (4-6). Please provide a higher-resolution file.

Regards

Author Response

Good afternoon,

           Thank you,for your time and also for the encouragements, hoping that we have accomplished the suggested requirements.

About the Title of the article, I suggest you to modify it and add the type of article.- Modified

-Please be sure to use only keywords accordingly to medical subject headings (Mesh word) for better indexing. diagnostics, stomatognathic structure, temporomandibular dysfunction,(TMDs) periodontal diseases, dental mobility, and oral rehabilitation . - Done

- The introduction section is very short and is needed to add other references to increase the quality of the manuscript, -modified

I suggest you some articles:  [DOI: 10.1080/08869634.2022.2137129] ; [DOI: 10.1155/2022/7091153] ;   [https://doi.org/10.3390/prosthesis4020025] ; [/doi/full/10.1080/08869634.2022.2126079]  added to references

-You need to review the grammar and English of your article, 

Figures are blurry (4-6). Please provide a higher-resolution file. =Modified

Round 2

Reviewer 1 Report (Previous Reviewer 1)

The authors responded to all the reviewer's comments. In my opinion, in its current form, the paper can be published. 

Author Response

Thank you very much .

This manuscript is a resubmission of an earlier submission. The following is a list of the peer review reports and author responses from that submission.

Round 1

Reviewer 1 Report

Jak widzę jest to resubmisja manuskryptu, który już recenzowałem. 

Część informacji, których wg mnie brakowało jest dodanych, ale wciąż są niewyjaśnione kwestie, które są poważnymi błędami.

1. title - it is pretty complicated but also confusing in terms of what data is in the abstract but also the body of the manuscript

2. TMD Pathology - what exactly does this mean? Or should it be TMJ Pathology? or maybe TMD etiopathology? 

3. the purpose of the study in the abstract - what does it mean at all - is different than in the body of the manuscript. 

4. the article needs proofreading by a native speaker. As it stands, some statements are confusing due to language errors. 

5. The number of patients should be systematized in the abstract, perhaps stating that only 75 out of 250 met the inclusion criteria. 

6 The introduction is a bit long. At some points, it looks like a discussion; some parts should be moved to the discussion. 

7. The objective is described in a way that makes it much more difficult to understand. What is the study's null hypothesis? 

8. why was the control group not as large as the study group? 

9. which diseases where excluded from the study group?

10. You wrote "Within the limitation of the study, our results provide genetic evidence that supports the possible causality of periodontal disease accounting for the host susceptibility to lose bone"

What are these limitations? 

11. what are the possible future directions of this study?

12. how can the results of the study be used in the daily practice of a dentist. 

13. it is very difficult to refer to the conclusions because the purpose was not specified. 

In my opinion, the article in its current form cannot be accepted for publication. I suggest the rejection of this article. 

Author Response

Good afternoon , and thank you once again for the review sesion article,hoping that this time will be a better conformation of us ,regarding the pertinent demands , we act as it  follows: 

  1. title - it is pretty complicated but also confusing in terms of what data is in the abstract but also the body of the manuscript

 Modified : PRACTICAL ASPECTS  REGARDING DENTAL MOBILITY PHENOMENA  IN THE EARLY DIAGNOSIS OF TEMPOROMANDIBULAR DISORDERS(TMD)

  1. TMD Pathology - what exactly does this mean? Or should it be TMJ Pathology? or maybe TMD etiopathology? 

We sincerely regret that the notions regarding the territory of temporomandibular joint damage were confusing. We would like to specify that we were concerned with temporomandibular joint disorders, found under the name of temporo-mandibular disorders, or temporomandibular joint pathology, an extremely complex clinical entity with a varied clinical picture that can affect the patient's quality of life in terms of the etiopathogenesis of temporomandibular pathology. This is a notion that includes the implication that dental mobility can have as a complication of periodontal disease with important impact on the modification of static and dynamic occlusion parameters.

  1. the purpose of the study in the abstract - what does it mean at all - is different than in the body of the manuscript. 

 Modified in conformity to the indication.

  1. the article needs proofreading by a native speaker. As it stands, some statements are confusing due to language errors. 

 done

  1. The number of patients should be systematized in the abstract, perhaps stating that only 75 out of 250 met the inclusion criteria. 

 done

6 The introduction is a bit long. At some points, it looks like a discussion; some parts should be moved to the discussion. 

Modified  the  introduction section .

  1. The objective is described in a way that makes it much more difficult to understand. What is the study's null hypothesis? 

The main objective of the study is to quantify the implications of the complications of periodontal pathology and dental mobility on the pathology of dysfunctional algo syndrome, a clinical entity with profound implications for the patient's quality of life.

The working hypothesis is based on the existence of correlative aspects between malocclusion and temporomandibular joint pathology as a general trajectory. We aim to verify, by quantifiable methods, the way dental mobility influences static and dynamic occlusion with the identification of consequences at TMJ level.

  1. why was the control group not as large as the study group? 

 Concerning the two study and control groups, we started from comparable sizes of the two groups, but it should be mentioned that a number of 10 patients withdrew from the control group during the study, this withdrawal being absolutely justified in the pandemic context that disrupted the healthcare activity. The maintenance of the two groups in the current number was supported by statistical theory which does not assume equal or larger sizes for the samples analyzed.

  1. which diseases where excluded from the study group?

We excluded uncooperative patients and those with periodontal pathology in the early stages, without significant bone damage and therefore without tooth mobility.

  1. You wrote "Within the limitation of the study, our results provide genetic evidence that supports the possible causality of periodontal disease accounting for the host susceptibility to lose bone"

We have removed this sentence from the article, the genetic aspects will be detailed in future approaches, the discussions will be much more extensive,

  1. what are the possible future directions of this study?

The future directions of study aim to extend the correlative aspects existing between the periodontal damage and the general condition of the patient and the existing morphological and functional changes in the temporomandibular joint territory. We aim to carry out a prognosis of the temporomandibular pathology in systemic context with different evolutionary trajectories of periodontal damage by means of a digital application.

  1. how can the results of the study be used in the daily practice of a dentist. 

Based on the fact that temporomandibular pathology has a negative impact on the patient's quality of life in a high proportion up to 70% according to some statistical studies, we aim the present article to be a warning signal of early interception of this somewhat dysfunctional syndrome as a loco-regional complication of periodontal pathology, especially dental mobility. Tooth mobility, a complication of periodontal pathology, disturbs static and dynamic occlusal ratios. Identification by specific periodontal assessment periodical examinations such as computed tomography or axiography is a relevant starting point for a precise diagnosis and a targeted therapeutic approach. It is extremely important for practitioners to identify the importance of the disturbance of occlusal ratios due to complications of periodontal disease and the negative effect on the temporomandibular joint. 

  1. it is very difficult to refer to the conclusions because the purpose was not specified. 

On the other hand, the direct causal link between malocclusion and temporomandibular disorders is extremely important for the diagnosis and therapy of temporomandibular pathology.

The correlation of specific signs of temporomandibular pathology in 64 patients correlated with the presence of tooth mobility and gingival recession in both samples is an eloquent plea for early initiation of therapy in the temporomandibular registry in patients with periodontal disease. 

Reviewer 2 Report

This revised version is equal to the original manuscript

Author Response

Good afternoon ,thank you for the review sessione, we try to do our best for having the best modality to expose what we figure it  in the research article . 
